# Acute Myeloid Leukemia Expresses a Specific Group of Olfactory Receptors

**DOI:** 10.3390/cancers15123073

**Published:** 2023-06-06

**Authors:** Gabriela D. A. Guardia, Rafaella G. Naressi, Vanessa C. Buzzato, Juliana B. da Costa, Ilana Zalcberg, Jordana Ramires, Bettina Malnic, Luciana M. Gutiyama, Pedro A. F. Galante

**Affiliations:** 1Centro de Oncologia Molecular, Hospital Sírio-Libanês, São Paulo 01308-060, SP, Brazil; gguardia@mochsl.org.br (G.D.A.G.); vbuzatto@mochsl.org.br (V.C.B.); 2Centro de Transplante de Medula Óssea, Instituto Nacional do Câncer, Rio de Janeiro 20230-130, RJ, Brazil; rafaellanaressi@usp.br (R.G.N.); juliana.bulchi@gmail.com (J.B.d.C.); zalcberg@inca.gov.br (I.Z.); jordanararagao@gmail.com (J.R.); 3Department of Biochemistry, University of São Paulo, São Paulo 05508-000, SP, Brazil; bmalnic@iq.usp.br

**Keywords:** olfactory receptors, cancer, acute myeloid leukemia, gene expression biomarkers

## Abstract

**Simple Summary:**

Acute myeloid leukemia (AML) is a type of cancer that affects blood cells and is the most common type of acute leukemia in adults. Despite advances in treatment, many patients with AML still face poor outcomes, including a high risk of relapse. Therefore, new ways to treat AML are needed. In this study, we have discovered a group of genes called olfactory receptors (ORs) that are highly expressed in AML cells. ORs are normally found in the nose and are responsible for detecting smells, and are not commonly found in other tissues or parts of the body. Here, we show that 19 ORs were predominantly found in AML cells and that the expression of these ORs could predict the prognosis of AML patients. In summary, we believe that these ORs could be further investigated to be used for diagnosis and as targets for new drugs to treat AML.

**Abstract:**

Acute myeloid leukemia (AML) is the most common form of acute leukemia in adults, with a 5-year overall survival rate of approximately 30%. Despite recent advances in therapeutic options, relapse remains the leading cause of death and poor survival outcomes. New drugs benefit specific small subgroups of patients with actionable therapeutic targets. Thus, finding new targets with greater applicability should be pursued. Olfactory receptors (ORs) are seven transmembrane G-protein coupled receptors preferentially expressed in sensory neurons with a critical role in recognizing odorant molecules. Recent studies have revealed ectopic expression and putative function of ORs in nonolfactory tissues and pathologies, including AML. Here, we investigated OR expression in 151 AML samples, 6400 samples of 15 other cancer types, and 11,200 samples of 51 types of healthy tissues. First, we identified 19 ORs with a distinct and major expression pattern in AML, which were experimentally validated by RT-PCR in an independent set of 13 AML samples, 13 healthy donors, and 8 leukemia cell lines. We also identified an OR signature with prognostic potential for AML patients. Finally, we found cancer-related genes coexpressed with the ORs in the AML samples. In summary, we conducted an extensive study to identify ORs that can be used as novel biomarkers for the diagnosis of AML and as potential drug targets.

## 1. Introduction

Acute myeloid leukemia (AML) is an aggressive cancer in hematopoietic progenitor cells that results in uncontrolled clonal proliferation of immature myeloid blast cells [1]. AML is the most common form of acute leukemia in adults, accounting for approximately one-third of all leukemias diagnosed, and the incidence increases with age. Approximately 20,380 new cases and 11,310 (31.7% 5-Year Relative Survival) deaths from AML are expected in 2023 in the United States alone [2]. In contrast to those for chronic myeloid and lymphocytic leukemia, AML survival curves have remained stagnant for many decades [3]. AML remains an infrequent but still lethal malignancy. Consequently, finding novel molecular targets in AML would be beneficial for the development of actionable therapeutic targets and potentially more effective drugs.

Olfactory receptors (ORs) constitute the largest family of G-protein coupled receptors in humans [4] and other mammals [5]. OR genes are highly transcribed (expressed) in olfactory neurons of the nasal olfactory epithelium, where each neuron expresses a single allele of one given OR gene, a remarkable mechanism of gene expression regulation [6,7]. OR proteins are comprised of seven transmembrane domains and are the key elements of the molecular machinery responsible for interacting with multiple odorant molecules and initiating a cascade of signal transduction events (neuronal responses), leading to smell perception [8,9]. OR proteins also participate in guiding olfactory sensory neuron axons to their glomerular targets [10] and participate in the regulation of their own choice and expression [11].

With the development of deep sequencing platforms and other molecular biology tools, there is a growing body of literature describing the ectopic expression of ORs, i.e., expression in extraolfactory tissues and cells [12]. The ectopic expression of ORs was first identified in germ tissues, specifically in sperm cells [13]. More recently, OR expression has been reported in dozens of healthy tissues from humans, mice, and other species [14,15]. Functional roles in health and pathological conditions have also been attributed to these ORs, including chemotaxis [16,17], melanocyte proliferation [18], energy metabolism [19], inflammation [20], and apoptosis [21]. Furthermore, a myriad of ORs with expression and putative functions in different types of cancers have been reported. For example, OR51E2 shows increased expression in prostate cancer cells [22] and a potential role in the origin and progression of melanoma [23]. OR51E1 shows increased expression in cancers of the small intestine, lung, and prostate [24,25,26]. OR7C1 and OR2C3 are markers for cancer-initiating cells in colon cancer and melanoma [27,28]. OR2AT4 expression is reported to be capable of inducing a proapoptotic process in leukemia cells [29]. Together with the fact that OR expression is predominantly absent in nonolfactory tissues and that ORs are likely to be expressed on the surface of these tumor cells, these findings suggest that ORs could be excellent targets for diagnoses and novel therapies in cancer [30,31,32].

Herein, we performed a deep and complete analysis of OR expression profiles in AML. By analyzing gene expression in more than 17,000 healthy and tumor samples, we found a set of ORs significantly overexpressed exclusively in AML. Furthermore, additional characteristics of these ORs were investigated, including gene subfamilies, genomic location, and methylation patterns, based on in-house and orthogonally validated cell lines and AML samples.

## 2. Materials and Methods

### 2.1. Gene Expression Profiles of ORs in Tumors and Healthy Tissues

First, to investigate the expression profiles of OR genes in acute myeloid leukemia (AML) and healthy tissues, we obtained gene read counts (RNA sequencing) of 151 AML patients from The Cancer Genome Atlas (TCGA) (https://portal.gdc.cancer.gov/ (accessed on 30 May 2023)) and 11,215 samples from 713 healthy individuals corresponding to 51 tissues from the Genotype-Tissue Expression consortium (GTEx v7; https://gtexportal.org/ (accessed on 30 May 2023)). Patient information can be found in Appendix A).

In addition, to explore the expression profiles of OR genes in 15 other types of tumors from TCGA, we also obtained gene read counts from 6439 patients. For further analyses, gene read counts from tumor and healthy samples were normalized to transcripts per million (TPM) using local R scripts.

### 2.2. Overexpression of OR Genes in AML Compared to Healthy Tissues

To compare the expression profiles (gene read counts) of OR genes between AML (TCGA) and each healthy tissue from GTEx, we applied DESeq2 [33]. We defined overexpressed OR genes as those with log_2_FoldChange > 1 and Benjamini–Hochberg adjusted *p*-value (FDR) < 0.05 in all comparisons.

### 2.3. Gene Expression Profiles of ORs in Leukemic Cell Lines

To further evaluate the expression of OR genes in leukemic cell lines, we obtained publicly available RNA-Seq data from multiple NCBI GEO datasets. In particular, we analyzed four acute myeloid leukemia cell lines (KASUMI-1, MOLM-13, NB-4, and OCI-AML-3; GEO identifiers: GSE111310 and GSE101821), in addition to two acute lymphoid leukemia (ALL) cell lines (MOLT-4 and REH; GEO identifiers: GSE103046 and GSE79871) and two chronic myeloid leukemia (CML) cell lines (K562 and KCL-22; GEO identifiers: GSE110229 and GSE62121). First, all sequencing reads were aligned against the human reference genome (hg38/GRCh38) using STAR [34], and only mapped reads with a mapping quality score (Q) ≥ 20 were selected with SAMtools [35]. Read counts per gene were obtained using HTSeq-count [36] and normalized to transcripts per million (TPM) using local R scripts.

### 2.4. Phylogenetic Analysis

To obtain the evolutionary phylogenetic tree for OR genes, we used Molecular Evolutionary Genetics Analysis (MEGA; version V7) software [37]. The phylogenetic tree was inferred by the maximum likelihood (ML) method based on the full-length nucleotide sequences of the 19 selected OR genes. Dendrograms were obtained by applying the neighbor-joining and BioNJ algorithms in a matrix of pairwise distances estimated using the maximum composite likelihood (MCL) approach. The codon positions included in the analyses were 1st, 2nd, 3rd, and noncoding. All positions containing gaps and missing data were eliminated. There were a total of 876 positions in the final dataset.

### 2.5. Coexpression Analyses in AML

To analyze the coexpression patterns among the 19 ORs and between the 19 ORs and ~20,000 protein-coding genes in AML samples, we used Spearman’s rank correlation. Genes with rho > 0.5 and *p*-value < 0.05 were considered positively correlated. Correlation analyses were performed using the R package corrplot (https://github.com/taiyun/corrplot (accessed on 30 May 2023)), with the hierarchical clustering of the correlation results.

### 2.6. DNA Methylation Profiles in AML

To investigate the DNA methylation profiles of AML samples, we downloaded Illumina Human Methylation 450K processed data from 140 AML patients available in the TCGA portal. Methylation patterns in the genomic regions containing the 19 ORs were compared to 10 random genomic regions obtained for each chromosome in all patients.

### 2.7. Samples from Patients and Healthy Individuals

We collected 20 mL of K2EDTA-anticoagulated blood from each CML and 5 mL of K2EDTA-anticoagulated bone marrow from each AML patient at diagnosis. Blood or bone marrow samples were obtained from healthy individuals. Each sample was mixed with 25 mL of cold RBC lysis buffer. After 10 min of incubation on ice, the samples were centrifuged at 500× *g* for 7 min at 4 °C. Subsequently, the supernatant was removed, and a second round of red cell lysis was performed as described previously. Finally, the cell pellet was washed with 15 mL PBS, and 5 × 10^6^ cells were used for each RNA extraction. The samples used for AML diagnosis for patients in this study were collected from 2014 to 2018. This study was approved by the Ethics Committee of the Brazilian National Cancer Institute with register number CAAE: 32965314.8.1001.5274.

### 2.8. Cell Line Culture

Eight human leukemia cell lines were evaluated. Four lineages were derived from AML patients: Kasumi-1 (ACC 220) with RUNX1-RUNX1T1 (AML1-ETO; t(8;21)); OCI-AML3 (ACC 582) with mutation type A in NPM1 and DNMT3A R822C mutation; NB-4 (ACC 207), from acute promyelocytic leukemia, with PML-RARa (t(15;17)); MOLM-13 (ACC 554) secondary AML FAB M5a after initial myelodysplastic syndromes, FLT3-ITD positive. Two other cell lines were obtained from CML patients: KCL-22 (ACC 519) with BCR-ABL1, t(9;22), e13-a2 (b2a2); K562 (ACC 10) with BCR-ABL1, t(9;22), e14-a2 (b3a2). The two remaining cell lines originated from ALL patients: MOLT-4 (ACC362) T-lymphoblastic leukemia cell line; REH (ACC 22) derived from B-cell precursor leukemia with ETV6-RUNX1 (TEL-AML1; t(12;21). All cell lines were cultured in RPMI-1640 (Gibco, Invitrogen Corporation, Carlsbad, CA, USA) supplemented with 10% (KCL-22, K562, REH, NB-4) or 20% (MOLT-4, OCI-AML3, MOLM-13, Kasumi-1) fetal bovine serum (FBS, Gibco) and 1% penicillin–streptomycin (Gibco) at 37 °C in a 5% CO_2_ atmosphere, according to the manufacturer’s instructions (DSMZ, German Collection of Microorganisms and Cell Cultures, Braunschweig, Germany). Exponentially growing cells were used in at least triplicate experiments. The cells were washed with phosphate-buffered saline (PBS), and 5 × 10^6^ cells were used for RNA extraction.

### 2.9. Dose-Response Curve

Kasumi-1 and MOLM-13 cell lines were plated at a density of 5 × 10^5^ cells per mL and exposed to different concentrations (200 μM, 100 μM, 50 μM, 10 μM, 0 μM) of odorants (Cinnamaldehyde and Eugenyl Acetate) diluted in medium RPMI-1640 (Gibco, Invitrogen Corporation, Carlsbad, CA, USA) supplemented with 10% FBS (Gibco) 0.1% dimethyl sulfoxide (DMSO, Sigma, St. Louis, MO, USA). Cells cultured in a medium with and without DMSO were used as the control. The cells were incubated with Trypan Blue Stain 0.4% (Gibco), and the living cells were counted every 24 h using a Neubauer counting chamber.

### 2.10. RNA Extraction and cDNA Synthesis

RNA was isolated from patients’ cells or from cell lines with TRIzol Reagent^®^ (Invitrogen, Waltham, MA, USA) according to the manufacturer’s instructions, except that the aqueous phase was recovered until the distance of approximately 0.5 cm from the interphase to avoid gDNA contamination, and eluted in 20 µL of deionized water. cDNA was synthesized from 1 µg of total RNA using random primers (Invitrogen) and Superscript II enzyme (Invitrogen) according to the manufacturer’s instructions. Each RNA was tested in the presence and absence of the Superscript II enzyme. After the reaction, the resulting cDNA was tested by ACTB PCR (primers listed in Appendix A) to exclude contamination by genomic DNA.

### 2.11. Two-Step End-Point RT–PCR

For OR52B6, OR2L5, OR5C1, OR2G2, OR4X2, OR13F1, and OR1L6, the first step of nested PCR was performed according to [38], with degenerate primers P26 and P27 (Appendix A), each at 2 µM, for 20 cycles. Next, we used a specific pair of primers for each OR (primer sequences are listed in Appendix A). The PCR was carried out with 1 µL of 1:10 first-step PCR dilution in deionized water, DNA polymerase GoTaq (Promega, Madison, WI, USA), according to the manufacturer’s instructions, and consisted of 95 °C for 5 min and 40 cycles of 95 °C for 30 s, 55 °C for 30 s, 72 °C for 1 min, and a final extension of 72 °C for 10 min. For the receptors OR52H1, OR13D1, and OR2AK2, we performed a two-step PCR using the same pair of specific primers for each receptor (listed in Appendix A) in the first and second PCRs, using the same thermal profile, with a change only in the number of cycles on each step. The first step of PCR was carried out with 1 µL of cDNA and DNA polymerase GoTaq (Promega) according to the manufacturer’s instructions and consisted of 95 °C for 5 min and 15 cycles of 95 °C for 30 s, 55 °C for 30 s, and 72 °C for 1 min and a final extension of 72 °C for 10 min. Next, the same protocol was performed using 1 µL of 1:500 first-step PCR diluted in deionized water, performed with the same protocol used in the first step for 40 cycles. This two-step PCR approach was used to increase the specific amplification of ORs. All the PCRs were performed in Veriti thermal cycler (Applied Biosystems, Waltham, MA, USA) emulated in 9600-mode. PCR products were detected through electrophoresis in 2% agarose gel stained with ethidium bromide according to a 100 bp DNA ladder (Promega). All the samples were repeated in at least two independent PCRs.

### 2.12. OR Signature with Prognostic Value

To evaluate the potential of the 19 OR genes in predicting the overall survival of AML patients, we obtained curated clinical data available for 140 patients from [39] and used Reboot [40] with a ridge regression model to obtain a minimal gene signature associated with prognosis. AML patients were stratified based on their expression signature scores (ROC established cutoff) to create Kaplan–Meier curves and to perform multivariable analyses with patients’ clinical data. To further evaluate the identified OR expression signature in an independent AML cohort, we obtained expression (RNA sequencing) and clinical data of 38 AML patients from the Beat AML consortium [41] (Appendix A) and applied the same methodology as described.

### 2.13. Functional Investigation of Coding Genes Coexpressed with ORs

To investigate the biological processes related to protein-coding genes that were coexpressed with OR genes from each OR subfamily, we performed Gene Ontology enrichment analyses using the ShinyGO [42] and REVIGO [43] web tools. Only GO terms with an FDR < 0.05 (hypergeometric test) were considered enriched in our analyses. To identify cancer genes reported in AML that were coexpressed with the 19 ORs, we used the COSMIC database as a reference [44]. We also identified transcription factors coexpressed with our set of ORs using a list of 1639 transcription factors obtained from [45]. Finally, coexpressed transcription factors were submitted to a protein–protein interaction analysis in Cytoscape [46] based on the STRING database [47]. Only transcription factors presenting at least two interactions with default scores > 0.4 were maintained in the network.

### 2.14. Ethics Approval and Consent to Participate

The use of samples from AML patients and healthy individuals was approved by the Ethics Committee of the Brazilian National Cancer Institute with register number CAAE: 32965314.8.1001.5274.

## 3. Results

### 3.1. Finding Ectopic Expression of ORs in AML

To examine the ectopic expression of olfactory receptors in AML, we used a comprehensive approach in which expression data from healthy and tumor samples, along with in-house validation and functional data, were integrated and thoroughly investigated (Figure 1A). The steps of the analysis are further described, and their results are presented in the next sections.

To find ORs that are over (or majorly) expressed in AML, we evaluated 151 samples from The Cancer Genome Atlas (TCGA). In this cohort, individuals present a median age of 56 years at diagnosis, with 55% of patients being male. Most individuals (58.3%) present a FAB classification ranging from M0 to M2, and median percentages of bone marrow and peripheral blood blasts are 72% and 39%, respectively (Appendix A). We compared the expression of human OR genes in these AML samples with that of 407 healthy whole blood samples obtained from the Genotype-Tissue Expression consortium (GTEx). Due to the difficulty in obtaining healthy bone marrow RNA-Seq, we used sequences from whole blood samples, which are commonly used as healthy controls for leukemia [48]. From 366 OR genes analyzed, 341 were expressed (normalized expression in transcripts per million (TPM) > 0.1) in at least one AML sample, with values up to 91.4 TPM in some samples, although only 127 OR genes had an expression higher than 1 TPM (Appendix A). Performing a differential expression analysis, we found 47 OR genes overexpressed in AML in comparison to healthy whole blood samples (log2foldChange > 1 and FDR < 0.05; Figure 1B, Appendix A). Next, we expanded our investigation of the expression of these ORs to multiple healthy tissue types. For this, we used data from GTEx, comprising approximately 11,000 healthy samples from 51 human tissues. Figure 1C confirms that the 47 ORs were highly expressed in AML, with rare (low or no) expression in healthy samples. As previously reported [13], among healthy tissues, the testis mostly showed ectopic expression of OR genes.

To further identify ORs that would exhibit greater enrichment in AML, we used more stringent criteria for selecting ORs overexpressed in AML compared to all analyzed healthy tissues. In this analysis, we identified a subset of 19 OR genes (log2foldChange > 1 and FDR < 0.05; Appendix A). These 19 ORs showed lower expression levels in the testis than the other OR genes (Figure 1C). Moreover, these 19 ORs were overexpressed in AML but not expressed or rarely expressed in all other healthy tissues, as shown in Figure 1D.

### 3.2. Dissecting the Expression of ORs in AML versus Healthy Tissues

To better investigate each of these 19 ORs, we further analyzed their expression profiles in AML and healthy tissue samples in TCGA and GTEx databases, respectively. First, we assessed the percentage of samples expressing the ORs qualitatively; for this analysis, samples with expressions over 0.1 TPM were considered positive. Figure 2A shows that most of them were expressed in at least 60% of AML samples, except for OR10A4, OR10A2, and OR10A5 (~45% of AML samples). Interestingly, five ORs (OR52B6, OR52H1, OR2AK2, OR13D1, and OR52K2) were expressed in approximately 90% of the AML samples. Conversely, these 19 ORs were expressed in less than 30% of healthy tissue samples, with many being expressed in less than 10% of the samples (Figure 2A). As expected, among the healthy tissues, OR expression was found in the testis and whole blood samples. Combined, they represented less than 5% of the samples expressing these ORs (Figure 2A). Moreover, most of the 19 ORs were expressed in only a small number of healthy tissue types (Figure 2B). For example, OR52B6 was expressed in the ovary and uterus; OR10A4, OR13D1, and OR2AK2 were expressed in testis samples; and OR2AE1 was expressed in several tissue types. However, these ORs were expressed only in specific tissues (e.g., OR52B6) and not in a high percentage of the samples corresponding to each tissue type, as we observed for AML samples (Figure 2B). Furthermore, the 19 ORs were not only expressed in a greater number of samples from AML patients, but their expression levels were also higher when compared to those in healthy tissues. To confirm this enrichment of expression in AML and low expression in healthy tissues, we comparatively quantified the expression levels of the 19 ORs. Figure 2C shows that they were expressed at higher levels in AML than in healthy tissues, including ORs such as OR2AE1, which was expressed in almost 30% of the analyzed healthy tissue types (Figure 2A). Moreover, we investigated these ORs in an independent AML cohort of 58 patients from the Beat AML consortium [41] and 11,215 normal samples from GTEx. Confirming our findings (Figure 2), we observed that all 19 ORs were expressed in AML samples and that, except for three ORs with low expression in both AML and healthy samples, all others (16; 84%) were overexpressed in AML (Appendix A). Together, these results reveal that our set of 19 ORs have an idiosyncratic expression pattern of being primarily expressed in AML samples with relatively high expression levels and also expressed in a few healthy samples but with low expression levels.

### 3.3. Features of the ORs Overexpressed in AML

To determine whether the 19 ORs share some features, we investigated their phylogenetic relationship, genomic locations, coexpression, and methylation patterns. First, we found that 14 (74%) of the ORs were grouped into only three olfactory receptor subfamilies (OR2, OR10, and OR52) (Figure 3A). Next, using Spearman’s rank pairwise correlation, we analyzed the association among the expression patterns of the 19 ORs. Interestingly, we confirmed that ORs closely related in the evolutionary phylogenetic tree, such as OR genes from subfamilies 2 and 10, exhibited positively correlated expression patterns (rho > 0.5 and *p*-value < 0.05) (Figure 3B), suggesting a potential functional redundancy. Since genes from the same OR subfamily have an origin based on tandem duplication and that coexpressed genes tend to be closer in the genome, we looked into the genomic locations of the ORs. Figure 3C shows that 12 (63%) ORs are in only two genomic loci: chromosome 1, q44 and chromosome 11, p15.4. Curiously, the former genomic region contains almost all ORs from subfamily 2, while the latter contains all ORs from subfamilies 10 and 52.

To gain insights into the controlling mechanisms possibly involved in the expression of these ORs, we investigated the DNA methylation patterns in the 140 TCGA AML samples. Since some tumors present a pronounced pattern of hypomethylation in some genomic regions [49], we hypothesized that it could explain the expression of these ORs, especially those that are closely located in the genome. Despite being relatively distinct, the genomic regions containing the ORs in AML samples exhibited no differences regarding their methylation patterns in comparison to other random genomic regions (Figure 3D). All of them had beta values ranging from 0.4 to 0.6, suggesting that they were methylated and that the controlling mechanisms of expression of these ORs are fine-tuned in AML and not a consequence of general hypomethylation. Thus, these results indicate that the 19 ORs have many similarities in terms of origin, genomic location, and coexpression. This also suggests that the mechanism controlling the expression of these different OR genes in the AML samples may be common, although it seems to be unrelated to their DNA methylation status.

### 3.4. Expression of the 19 OR Genes in Other Cancer Types and Experimental Validation in AML Samples and Cell Lines

To ascertain that the expression of the 19 ORs is preferential to leukemia and not to cancer in general, we investigated their expression in 6000 tumor samples from 15 other cancer types available in the TCGA database: prostate, breast, rectum adenocarcinoma, glioma, thyroid, uterus, lung, pancreas, colon, bladder, liver, stomach, glioblastoma, and skin. Figure 4A shows a clear pattern of higher expression of the 19 ORs in AML than in the other cancer types: the median expression of the ORs was 0.44 TPM in AML and 0 in all other cancer types. Moreover, we performed a principal component analysis using the expression of the 19 ORs in AML and in the 15 other cancers. Figure 4B shows that AML is totally distinct from other cancers when we consider the expression of these 19 ORs. Together, these results reveal the potential of using these 19 ORs as biomarkers for AML since they have a distinct expression pattern in this tumor type in comparison to healthy tissues and various types of cancer. We also investigated the expression pattern of each OR individually (Appendix A). To illustrate their potential, Figure 4C shows the expression profiles of OR2G2 and OR13D1, which are enriched in AML compared to other tumor types. Thus, together, these results show that our set of 19 ORs has an AML-specific expression pattern even in comparison to 6000 samples from 15 other cancer types.

Next, we used an orthogonal methodology (RT–PCR) to confirm the expression of some candidates. Considering the low level of expression of ORs in AML samples, we developed a two-step PCR model. We have designed a pair of primers for each one of the 19 selected ORs, plus 1 OR not expressed in AML patients according to TCGA RNA-Seq analysis (OR4X2) and for OR51B5, previously described in the literature to be expressed in AML patients (Appendix A, [50]). All PCR products were sequenced and aligned to the reference sequence of the expected target OR. Each reverse and forward sequence of each OR was aligned to the human genome using BLAST. Each OR was identified with at least 92% of identity (Appendix A). For some alignments, some other targets were listed by BLAST, though the expected OR was the target with a higher score and longer identity. Every RT-PCR was performed in samples converted in the presence and absence of transcriptase reverse enzyme to detect genomic contamination. From our set of 19 ORs, due to the scarcity of RNA from patients’ diagnostic samples, we randomly selected nine genes and investigated their expression levels in a different subset of bone marrow or peripheral blood from AML patients and healthy donors: OR52H1, OR52B6, OR13D1, OR2AK2, OR2L5, OR5C1, OR2G2, OR13F1 and OR1L6, plus the nonexpression control OR4x2. As expected, each of the ORs was expressed in at least three samples, and most of them were expressed in six or more AML samples (Figure 4D; Appendix A). We observed that some of the ORs positive in donor samples by RT-PCR were the ORs expressed (TPM > 0) in healthy tissues from GTEx: OR52K2 (0.08854 TPM), OR52K1 (0.07545 TPM), OR52H1 (0.08778 TPM) and OR52B6 (0.07217 TPM). In contrast, the AML-specific targets according to RT-PCR (Figure 4D; Appendix A), such as OR2G2, OR1L6, OR13D1, and OR13F1, are the ORs with lower expression quantification in GTEx. This result demonstrates that the RT-PCR technique is in line with RNA Seq databases, and it is very sensitive and specific. The negative control OR4X2 was not expressed in any AML or healthy donor sample by RT-PCR, in agreement with AML-TCGA RNA Seq analysis.

As mentioned, the AML sample from a patient at the diagnostic is very limited; therefore, we extended our investigation to leukemic cell lines from AML, acute lymphoid leukemia (ALL), and chronic myeloid leukemia (CML). We used our RT-PCR in parallel to RNA Seq from each cell line downloaded from NCBI GEO databases. As AML patients, none of the tested cell lines were positive for OR4X2, while we confirmed the OR51B5 expression in our cell line K562 [50]. The group of ORs found to be expressed in each cell line in the NCBI GEO database was not entirely confirmed through RT-PCR (Appendix A). We found that 89% (17) of ORs had their expression confirmed in four cell lines from AML (Appendix A). Curiously, in line with our findings in AML patient samples, all cell lines exhibited expression of more than one OR despite being a homogeneous pool of cells, just the opposite of an AML patient’s sample at diagnosis. This data suggests that the regulation of the expression of ORs in AML blasts is different from the regulation of ORs expression in the OSNs, which follows the one OR per neuron rule. The expression of these ORs was also confirmed in ALL and CML cell lines (Appendix A).

### 3.5. An OR Expression Signature with Prognostic Potential

Since there is a high level of genetic heterogeneity in AML and, as expected, our set of 19 ORs has a distinctive expression profile in AML patients, we evaluated their potential use to predict overall survival rates in AML patients treated with a classical chemotherapy regimen (7 + 3). Rather than considering the expression levels of ORs individually, we used Reboot [40], an algorithm that seeks a combination of genes whose expression profiles have prognostic value and performs multivariable analyses with patients’ clinical data. By using Reboot and clinical data available from 140 AML patients in TCGA, we found a minimal OR expression signature of 7 ORs capable of categorizing AML patients into groups with longer (median OS: 12 months) and shorter (median OS: 8.1 months) overall survival (Kaplan–Meier curve; *p*-value = 0.038) based on their expression signature scores (Figure 5A and Appendix A). Next, we tested this OR expression signature in an independent AML cohort from the Beat AML consortium [41]. Notably, we also confirmed the prognostic value of the OR signature in this independent AML cohort (Figure 5B), in which AML patients from groups with longer and shorter overall survival (OS) times had median OS times of 7.6 and 6.5 months, respectively. We also performed a multivariate analysis considering some of the prognostic factors postulated by WHO guidelines [51], such as the patient’s white blood cell (WBC) count and mutational status of the FLT3 and NPM1 genes. In line with the literature [51,52], higher WBC and mutated FLT3 correlated with a poor prognosis, and mutated NPM1 with a good prognosis. However, none of the covariates, except the OR signature, significantly contributed to patient categorization (Figure 5B, highlighted in the table).

Next, we analyzed the OR genes comprising this prognostic signature. Figure 5C shows the genes and their expression levels in AML samples from patients with longer or shorter overall survival. Interestingly, the patients with a shorter overall survival exhibited a higher expression of three ORs: OR2G2, OR2AE1, and OR10A2, Figure 5C (boxplot, right side). Curiously, we noticed that these three ORs (OR2G2, OR2AE1, OR10A2) are from distinct genomic loci (Figure 3C), suggesting distinct (but synergic) expression control. Two of them (OR2G2 and OR2AE1) that are from the same olfactory receptor subfamily (OR2) are in distinct branches in the phylogenetic tree (Figure 3A), indicating a low sequence identity. Additionally, as shown above in Figure 4C, according to TCGA data, OR2G2 was overexpressed in AML in comparison to other cancers. Figure 4D shows that OR2G2 was expressed in 8 out of 13 (62%) AML samples and in only 2 out of 14 (14%) healthy control samples, with a clear higher frequency in tumor samples.

### 3.6. A Functional Investigation of Genes Coexpressed with the 19 ORs

Cancer cells usually acquire a multitude of alterations in gene expression resulting in synergic changes in key cellular pathways to promote tumor growth [53,54]. It is common to observe the coexpression of genes from these pathways and tight protein–protein interaction networks in large-scale analyses. Additionally, since ORs are cellular receptors that can interact with external molecules and initiate an intracellular cascade of signal transduction events [55,56], we hypothesized that the 19 ORs could be coexpressed with other genes with which they act synergistically in AML tumors.

To test this hypothesis, in AML samples, we searched for co-expression among the set of 19 ORs and all other (~20,000) known protein-coding genes. Using Spearman’s correlation rank test, we selected 1107 genes showing a positive expression correlation (rho > 0.5; *p*-value < 0.05) with at least one of the 19 OR genes (Appendix A). To obtain a broad view of the biological functions carried out by these genes, we performed a Gene Ontology analysis (Figure 6A). Curiously, in comparison to all genes, the ORs’ coexpressed genes were enriched in biological processes (FDR < 0.05) already described and important in carcinogenesis, such as superoxide anion generation [57], phosphatidylcholine metabolism [58], and regulation of cytokine production [59], to name only a few (Figure 6A).

Next, we investigated the coexpression of the set of ORs with transcription factors (TFs), an important class of regulators frequently altered in numerous cancer types [60]. We found an enrichment of TFs coexpressed with ORs pertaining to three OR subfamilies: OR2, OR13, and OR52 (Fisher’s test, *p* value< 0.05; Appendix A). We then performed a protein–protein interaction analysis, which resulted in a well-connected network of TFs (Figure 6B). Notably, well-known cancer-related TFs emerged, such as MYCN [61], GATA2 [62], and ESR1 [63]. Interestingly, all of them had already been reported to be altered in AML, MYCN [64], GATA2 [65], and ESR1 [66]. Finally, we directly investigated the coexpression with the set of ORs of cancer genes already reported in AML according to the COSMIC database [44], and most (77%) of the coexpressed genes have proto-oncogene functions (Figure 6C). Overall, this large-scale functional investigation of genes coexpressed with our set of 19 ORs revealed a fruitful list of genes, some of which are clearly involved in cancer and AML, while others are involved in key pathways to carcinogenesis.

### 3.7. The Growth of AML Cell Lines Expressing ORs Are Affected in a Dose-Response Manner by Treatment with Odorant Molecules

Previous data from the literature [21,22,67,68,69,70,71] have demonstrated that the activation of different ORs expressed in cancer cell lines may influence tumor growth, affecting cell proliferation and, in some cases, cell death. Therefore, we treated two of the analyzed AML cell lines with odorants (chemical compounds that can be detected by ORs) and evaluated if the total number of living cells was affected. From our 19 ORs, only two (OR2G2 and OR52B6) had been deorphanized (i.e., with odorants identified). However, OR2G2 was not expressed in any of the AML cell lines, and OR52B6 only had a positive expression both from the RNA-Seq analysis and PCR in an ALL cell line (MOLT-4) (Appendix A). For this reason, knowing that ORs with phylogenetic proximity tend to respond similarly to the same odorants [72,73], we selected two AML cell lines known to express OR2B2, MOLM-13 (AML FAB M5a cell line, confirmed both in RNA-Seq and rt-PCR analysis) and Kasumi-1 (AML FAB M2 cell line, negative in RNA-Seq analysis but positive in rt-PCR) and treated with eugenyl acetate [74] and cinnamaldehyde [75,76], two ligands of OR2B11, which belongs to the same phylogenetic family of OR2B2. Remarkably, the number of viable cells was significantly smaller for the MOLM-13 cell line when incubated with cinnamaldehyde (*p*-value: 1.63 × 10^−5^) and for cells incubated with eugenyl acetate (*p*-value: 4.93 × 10^−3^) in concentrations of 200 µM after 5 days of treatment. The results obtained with Kasumi-1 were similar in the presence of cinnamaldehyde (*p*-value: 7.58 × 10^−6^) and eugenyl acetate (*p*-value: 9.12 × 10^−3^) (Figure 7). These dose-response results indicate that possibly: (i) the OR2B2 proteins are located in cell membranes and (ii) the presence of odorant molecules may affect the tumor growth in vitro in a dose-response manner.

## 4. Discussion

Ectopic expression of ORs has been reported since the early 1990s [13]. In cancer, OR expression has been identified in non-small-cell lung cancer, gastrointestinal, neuroendocrine, hepatocellular, and prostate carcinomas, among others [32,77,78,79,80,81,82]. In some cases, the activation of these ORs led to apoptosis, inhibition of cell migration and proliferation, differentiation, and resistance to drugs [18,21,32,77,82,83]. Here, we performed an original and complete investigation of the ectopic expression of olfactory receptors (ORs) in acute myeloid leukemia (AML) NGS publicly available databases, patients, and cell lines. We found a small set of 19 ORs primarily expressed in AML, which exhibited rare or low expression in healthy tissues and other cancer types. Additionally, through several systematic analyses, we showed that these ORs had a distinct expression pattern, a significant predictive value for overall survival in AML patients, and were coexpressed with key cancer genes. We also highlight the potential of ORs to be used as novel biomarkers for diagnosis and drug targets, not only in AML but also in other types of cancer [30,31,32].

First, by using a myriad of gene expression datasets (RNA sequencing) and a well-tuned set of computational pipelines and parameters, we found 19 ORs overexpressed in the AML samples in comparison to whole blood healthy samples and 51 other tissues (approximately 11,500 healthy samples) (Figure 1). Next, to obtain the full expression profile of the ORs, we analyzed their expression individually in healthy samples. Most of the ORs were expressed in less than 5% of healthy tissue samples, and only three ORs (OR2AE1, OR52B6, and OR2L3) were expressed in a range of 15% to 28% of samples (Figure 2). As expected, only the testis expressed a distinct set of ORs (but at lower levels). The testis was one of the first tissues reported to have an ectopic expression of ORs, and it is a tissue known for showing broad expression of ORs and other genes in general [13]. Other tissues also exhibited expression of some ORs; however, the expression was not homogeneous throughout all samples and was at a much lower level in most samples. On the other hand, 84% (16) of the 19 ORs were expressed in at least 60% of AML samples (Figure 2), and 4 ORs were expressed in more than 90% of the AML samples. CD33 is a canonical AML marker broadly expressed in AML blast and stem cells [84,85,86,87]. Despite many attempts, the use of CD33 as a therapeutic target is now restricted to one single antibody-conjugated drug named Gemtuzumab Ozogamicin. Randomized trials show that the clinical benefits are predominantly seen in favorable and intermediate-risk cytogenetics patients. High toxicity from thrombocytopenia and hepatotoxicity with an increased risk of veno-occlusive disease often occurs [88,89,90]. Moreover, similar to the 19 selected ORs (Figure 2), CD33 is not expressed outside the hematopoietic system. Considering that the lack of cancer-restricted surface markers expressed in the majority of AML subtypes is one of many barriers to improving AML management, the predominance and high frequency of expression of the ORs in AML samples are remarkable. Our results also show that the 19 ORs are expressed at low levels in healthy tissues but at high levels in AML (Figure 2). In the future, it would be of great interest to investigate the expression of these 19 ORs through the hematopoiesis process since we have not evaluated their expression in normal myeloid progenitor cells. CD33, for example, is expressed in some myeloid cell types from the normal hematopoietic system, which represents the main reason for treatment toxicity. Interestingly, Maiga et al. [48] investigated GPCR expression in 148 AML samples and 12 samples of CD34^+^ cells from healthy cord blood, blood, and bone marrow. All 19 ORs except OR2L5 were expressed in at least one AML sample at >1 RPKM, and their expression in normal CD34^+^ cells was very low or absent. Previous studies have already reported the expression of ORs in human blood leukocytes, but these ORs are different from the ones identified in this study, and all of them were Class I ORs [91], whereas the 19 ORs are from both Classes I and II. Thus, for the first time, we report a set of 19 ORs primarily expressed in AML samples but with no or rare expression in healthy tissues (including blood cells). These ORs (individually or in combination) can potentially be used as molecular biomarkers for AML. These markers would especially benefit patients with normal karyotypes and without genetic targets.

Five of the 19 ORs are located at genomic position 1q44, and the other 7 ORs are located at 11p15.4. Both *loci* are known to contain high concentrations of OR genes [4]. It is known that ORs within the same subfamily and closer in chromosomal location have a higher chance of being coexpressed, which was confirmed by our coexpression and phylogenetic data (Figure 3). Moreover, considering that DNA hypomethylation is a common feature in cancer genomes and that it might have effects on gene expression [49], we investigated the DNA methylation profiles of the regions of the 19 ORs in AML samples. No differences were observed in comparison to other sets of randomly selected genomic regions (Figure 3), suggesting that the expression of these ORs is not a subproduct of global DNA hypomethylation but rather a consequence of a regulated process of gene expression in AML.

ORs constitute the major GPCR family in mammals, and GPCRs make up 50% of all drugs sold in the USA [27,80]. Moreover, GPCRs have been demonstrated to have oncogenic potential when overexpressed in cells, as in thyroid adenomas, colon adenomas, and carcinomas, among other cancers [80]. OR activation can trigger many physiological processes in the cell, even if initiating alternative signaling pathways instead of the canonical one. It has been published that ORs in cancer when activated, can increase the Ca^2+^ concentration in the cytosol [92], phosphorylate p38 and Erk1/2 MAPK [21], stimulate the cAMP/PKA pathway and EGFR and AKT signaling cascades, and increase cell proliferation, angiogenesis, cell cycle arrest, cell survival, and apoptosis [71,79,93]. In acute lymphoblastic leukemia, mutations in OR2C3 have been associated with the development of the disease during childhood [94]. Leukemic cells are known for having abnormal signaling and constitutive activation in the PI3K-Akt-mTOR pathway, which is a very important pathway for hematopoiesis [95]. Searching for PI3K-Akt-mTOR pathway differences in AML patients, the authors [95] reported differential expression of ORs and other proteins involved in GPCR signaling. Interestingly, the expression of ORs was associated with tumor heterogeneity in AML. Since the receptor controlling the PI3K pathway could be a GPCR, this pathway may be activated by an OR; thus, they have speculated about the possibility of ORs being metabolic sensors in the bone marrow microenvironment in AML [95].

Expression of the OR genes both in AML patient samples and in cell lines was confirmed by RT–PCR. Interestingly, the investigated leukemia cell lines expressed two or more ORs, as previously reported [29,50] for a human cell line derived from metastasis of a pancreatic carcinoma (BON cell line) analyzed by a single cell RT-PCR. On the other hand, another previous study using olfactory sensory neuron cell lines reported that clonal cells contain subsets of cells expressing unique ORs (following the canonical rule of one neuron, one OR). Nevertheless, it is still unclear whether the expression of ORs in cancer cells does not follow the canonical rule. Cytogenetic abnormalities are widely used for risk classification of AML patients for assessing treatment response, although occasionally controversial for intermediate-risk patients [96]. Here, we explored an alternative approach based on gene expression data, which focused on a set of ORs primarily expressed in AML. We used a straightforward pipeline [40] to evaluate the value of OR expression to predict overall survival in AML patients submitted to a 7 + 3 chemotherapy regimen. We found a set of seven ORs (OR2G2, OR2AE1, OR10A2, OR10A4, OR5C1, OR13F1, and OR1L6) whose expression levels have prognostic value. Notably, we also validated this OR signature in an independent cohort of AML patients (Beat AML consortium [41]), confirming its robustness. We also observed that the group of patients with a shorter overall survival showed significantly higher (*p*-value < 0.05) expression of three out of these seven ORs (OR2G2, OR2AE1, OR10A2), while the group of patients with a longer overall survival showed higher expression of the other four ORs (OR10A4, OR5C1, OR13F1, and OR1L6), Figure 5. To date, no specific role has been reported for these ORs in cancer [97,98,99].

Additional research is required to determine at which point in the clonal evolution of AML the ORs are expressed, whether the selected ORs are expressed exclusively in leukemic cells or also in preleukemic stages and whether the ORs could help monitor measurable residual disease (MRD) and guide treatment response for AML. Additionally, the presence of these receptors in the cell membrane could allow them to be used as targets for therapies with humanized monoclonal antibodies or CAR-T cell therapy suitable for AML [30,31,100]. Notably, we demonstrate that treatment with odorant molecules in a dose-response manner (Figure 7) can affect the growth of AML cell lines expressing ORs, which opens avenues for investigating odorants as novel drugs to treat tumors expressing ORs.

## 5. Conclusions

In conclusion, we describe a systematic and complete investigation of the ectopic expression of olfactory receptors in acute myeloid leukemia. We found a set of 19 ORs with higher and preferential expression in AML samples, and a subset of these ORs could be used to categorize patients into two groups with significantly different overall survival rates. In the AML samples, the 19 ORs were coexpressed with key cancer genes. Since ORs are G protein-coupled receptors located in the cell membrane, they have a clear potential to be used as molecular markers for diagnosis and targets for the development of novel therapies (e.g., monoclonal antibodies) for AML, a still deadly type of cancer.

## Figures and Tables

**Figure 1 cancers-15-03073-f001:**
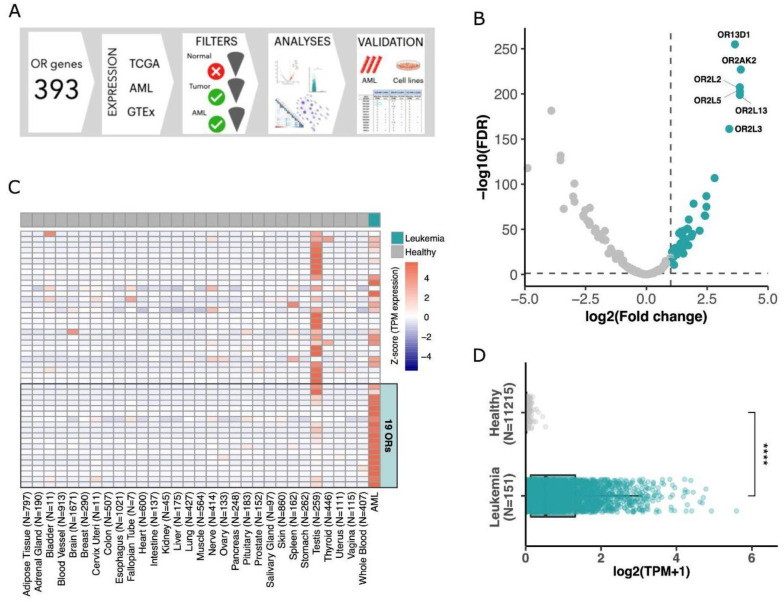
Finding olfactory receptors majorly expressed in AML. (**A**) Schematic view of the approach used to investigate OR gene expression in AML. (**B**) Volcano plot showing 47 ORs (green dots) differentially expressed in AML in comparison to healthy blood samples. (**C**) Expression of 47 ORs in healthy tissues and in AML tumors with a focus on the top 19 ORs overexpressed in AML. (**D**) Expression of the top 19 overexpressed ORs in AML (151 samples) and healthy samples (11,215 samples from 51 tissues) (Wilcoxon test; **** *p*-value < 0.0001).

**Figure 2 cancers-15-03073-f002:**
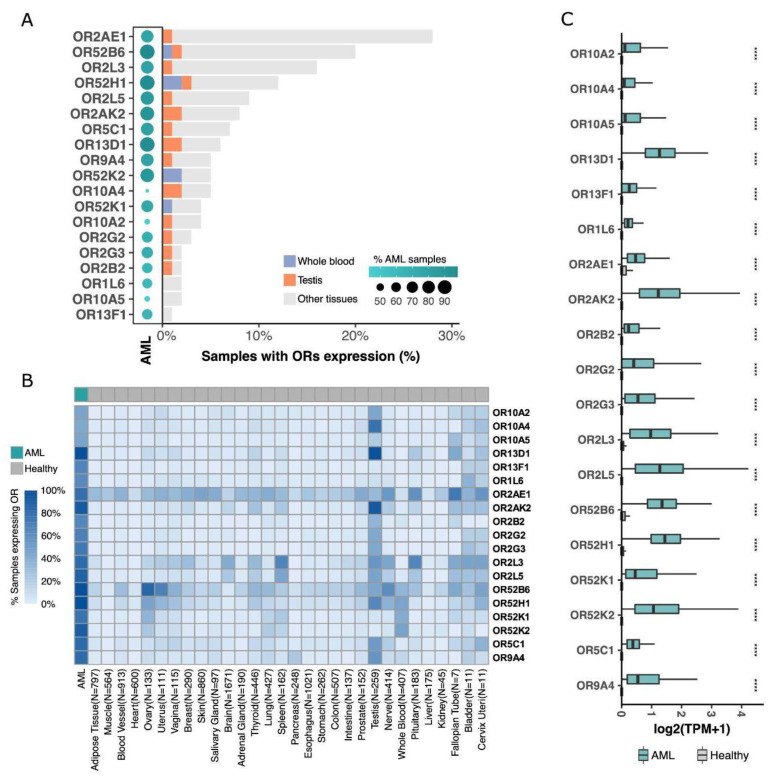
19 ORs are overexpressed in AML, with low or no expression in healthy samples from 51 distinct tissue types. (**A**) OR distribution over TCGA AML samples, whole blood, testis, and other healthy tissues (GTEx samples). (**B**) Percentage of samples expressing ORs, grouped by tissue type. (**C**) Comparative expression profiles of 19 ORs in AML and in healthy tissues (Wilcoxon tests; **** *p* < 0.0001).

**Figure 3 cancers-15-03073-f003:**
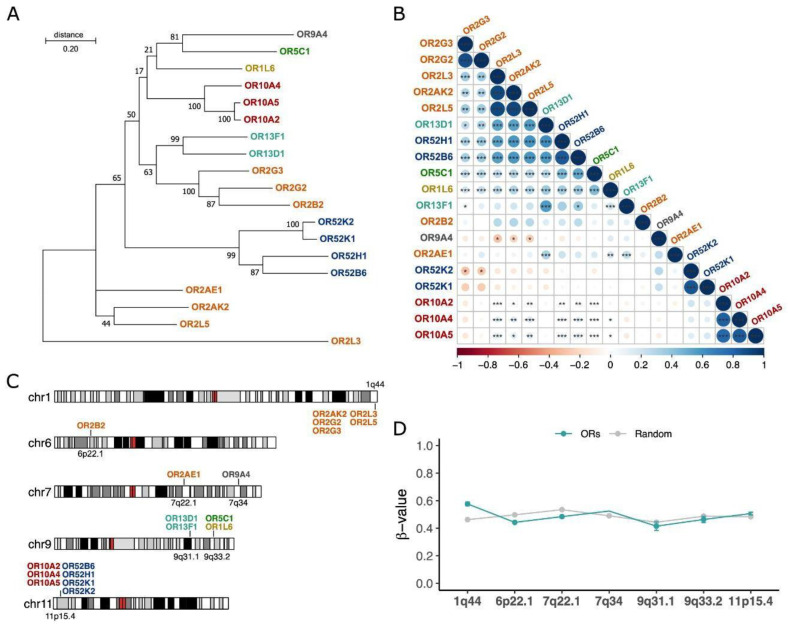
ORs overexpressed in AML belong to 7 gene subfamilies, are highly correlated in terms of expression, and are located in a few genomic loci. ORs are colored by their subfamilies. (**A**) Evolutionary phylogenetic three of the 19 ORs inferred by maximum likelihood based on their full-length nucleotide sequences. (**B**) Coexpression patterns of ORs in AML samples (Spearman’s rank test; * *p* value < 0.05; ** *p* value  <  0.01; *** *p*-value  <  0.001). The color bar indicates correlation coefficients ranging from −1 (red; negatively correlated) to 1 (blue; positively correlated). (**C**) Genomic locations of ORs are colored based on their subfamilies. (**D**) Methylation patterns in the genomic regions of the 19 ORs compared to random genomic regions of each chromosome of 140 AML patients from the TCGA database.

**Figure 4 cancers-15-03073-f004:**
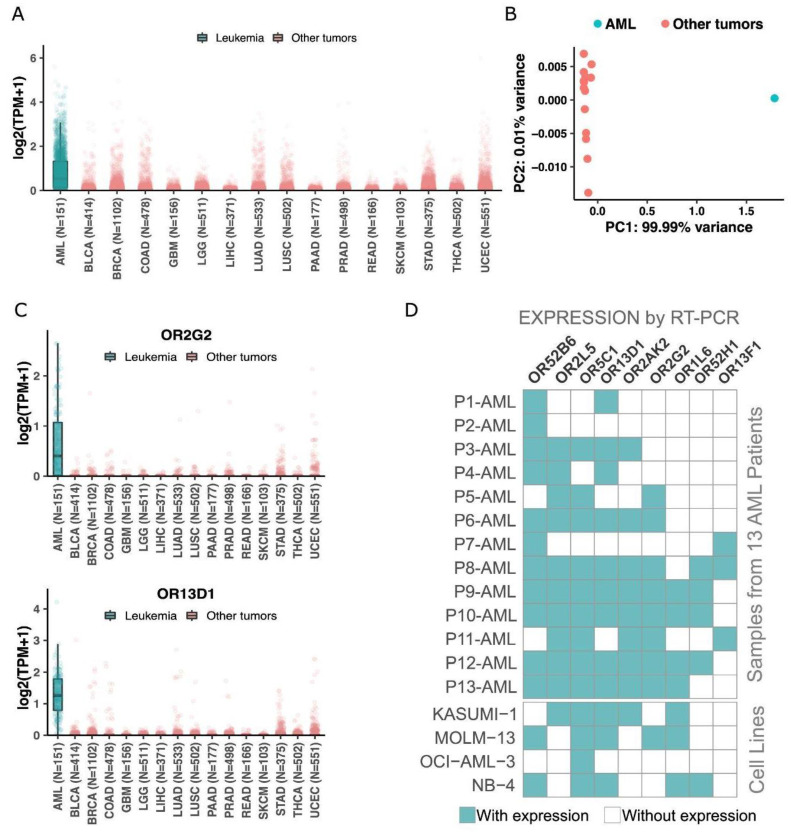
The 19 ORs are preferentially expressed in AML when compared to other cancer types. (**A**) Expression of the 19 ORs in AML and in other 6000 samples from 15 cancer types. (**B**) Principal component analysis of the 19 ORs in AML and 15 other cancer types. (**C**) Expression of two ORs in AML and other cancers. (**D**) Confirmation of expression by end-point RT–PCR of 9 randomly selected ORs in an independent AML patient cohort and AML cell lines.

**Figure 5 cancers-15-03073-f005:**
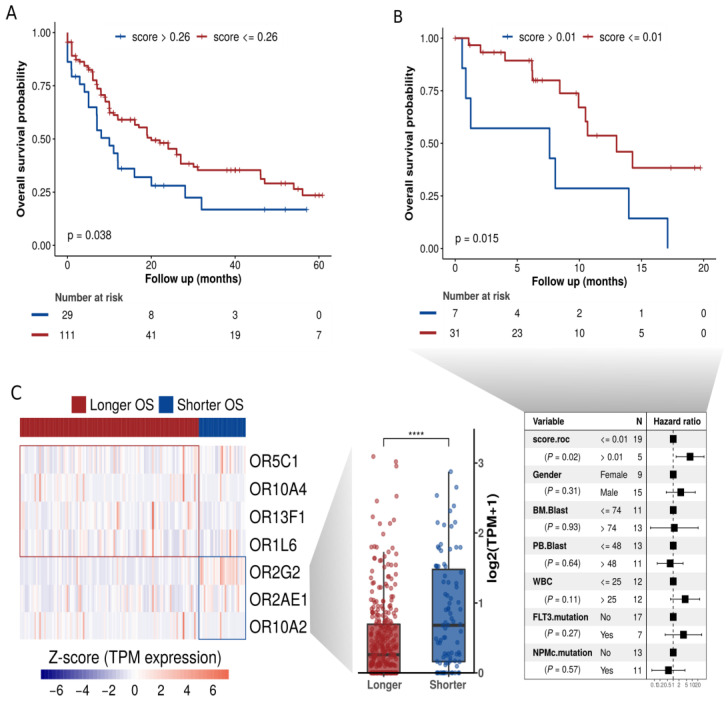
OR expression signature at diagnosis significantly correlates to overall survival in AML. (**A**) Overall survival curve based on clinical data from 140 TCGA patients and combined expression of seven ORs (signature scores). (**B**) Orthogonal validation of our OR signature in clinical data from 38 patients of the independent cohort BEAT-AML. (**C**) ORs present in the signature and their expression levels in patients with longer or shorter overall survival. **** *p* < 0.0001.

**Figure 6 cancers-15-03073-f006:**
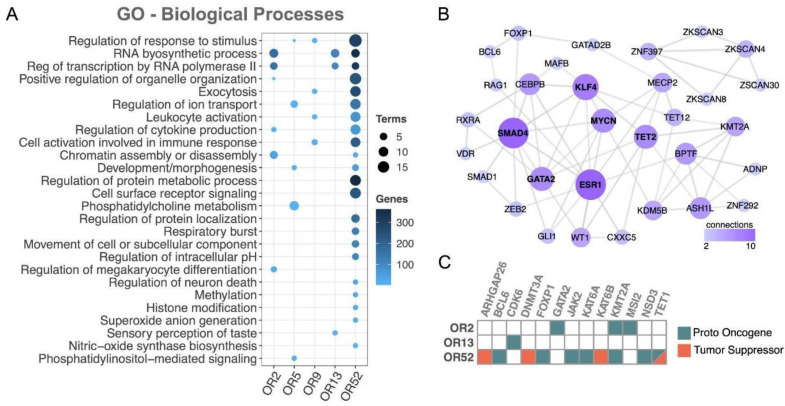
Functional investigation between coexpressed protein-coding genes and the 19 ORs revealed cancer genes and key cancer pathways. (**A**) Gene ontology (biological process) of the set of coexpressed coding genes (FDR < 0.05). (**B**) Network of protein–protein interactions among the set of coexpressed transcription factors. (**C**) Cancer genes (proto oncogenes or tumor suppressor) coexpressed with ORs. To simplify, ORs from subfamily 2 (OR2AE1, OR2AK2, OR2B2, OR2G2, OR2G3, OR2L3, and OR2L5) are labeled as OR2, ORs from subfamily 5 (OR5C1) are labeled as OR5, ORs from subfamily 9 (OR9A4) are labeled as OR9, ORs from subfamily 13 (OR13D1 and OR13F1) are labeled as OR13, and ORs from subfamily 52 (OR52B6, OR52H1, OR52K1, and OR52K2) are labeled as OR52. Only OR subfamilies showing at least one coexpressed protein-coding gene with Gene Ontology, PPI, or matching to cancer genes were considered.

**Figure 7 cancers-15-03073-f007:**
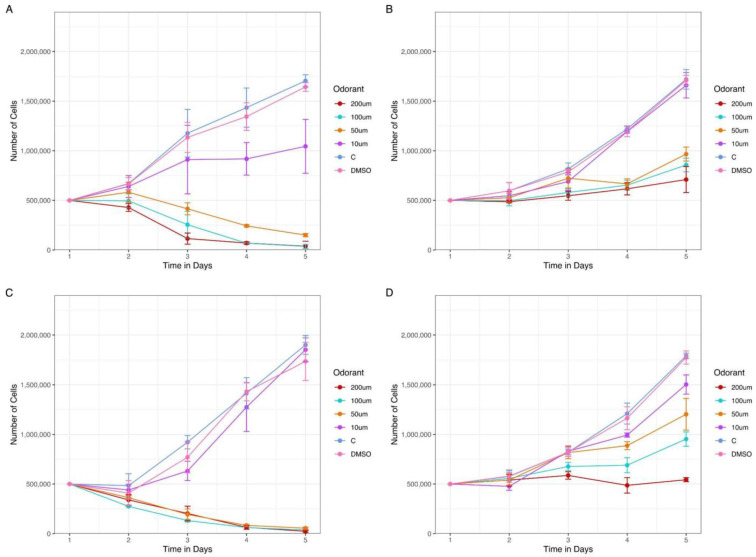
The presence of odorants in AML cell lines culture reduces the number of expected living cells in a dose-response manner. Two AML cell lines expressing OR2B2, Kasumi-1 and MOLM-13, were treated with increasing concentrations of two odorants, cinnamaldehyde and eugenyl acetate. Kasumi-1 cell line in the presence of (**A**) cinnamaldehyde and (**B**) eugenyl acetate, both odorants ranging from 10 to 200 µM. Molm-13 cell line in the presence of (**C**) cinnamaldehyde and (**D**) eugenyl acetate. We observe an initial difference in living cell count since the second day of treatment. The dose-dependent response was more evident from day 3 for both cell lines with the two odorants.

## Data Availability

Publicly available datasets were analyzed in this study. This data can be found at TCGA (https://portal.gdc.cancer.gov/ (accessed on 30 May 2023)), GTEx v7; https://gtexportal.org/ (accessed on 30 May 2023)), and GEO (https://www.ncbi.nlm.nih.gov/geo/ (accessed on 30 May 2023)) under accession identifiers GSE111310, GSE101821, GSE103046, GSE79871, GSE110229 and GSE62121.

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
