# Peer review of "Acute Myeloid Leukemia Expresses a Specific Group of Olfactory Receptors"

_cancers, 2023, doi:10.3390/cancers15123073_

Round 1

Reviewer 1 Report

This is a novel study that has identified that some specific olfactory receptors are expressed in acute myeloid leukaemia.    The study has initially used bioinformatic analysis of large publically available data sets to identify a smaller subset of OR.   These are subsequently reduced to eight to form the basis of a prognostic model.

Overall the study has been well done and there are only a few issues that the authors should consider.    

I am not sure that the methylation analysis is relevant for this study, it doesn't add to much the results and it may be that an extended methylation study would be more beneficial.

The rationale behind the phylogenetic tree was not clear until the drug exposure section.   Perhaps the authors could comment about the redundancy of the ORs and can they compensate for others?

The authors in Figure 6C showed some correlation with some relevant cancer genes, however, it would be useful to show if the differences in outcome shown were related to any specific gene or groups of genes (methylation related for example)

Author Response

Reviewer #1

Reviewer's points (in black)

Authors' response (in blue)

Comments and Suggestions for Authors

This is a novel study that has identified that some specific olfactory receptors are expressed in acute myeloid leukaemia.    The study has initially used bioinformatic analysis of large publically available data sets to identify a smaller subset of OR.   These are subsequently reduced to eight to form the basis of a prognostic model.

Overall the study has been well done and there are only a few issues that the authors should consider.    

We appreciate the time and effort you have dedicated to reviewing our manuscript. Your positive evaluation of our work is truly gratifying, and we are thrilled that you found our research valuable and well-presented. Your feedback and suggestions have been helpful in refining our manuscript.

I am not sure that the methylation analysis is relevant for this study, it doesn't add to much the results and it may be that an extended methylation study would be more beneficial.

We thank the reviewer for raising this point. However, we respectfully disagree that "Methylation analysis" does not contribute significantly to our results. As mentioned in the manuscript, DNA methylation is a common characteristic in cancer genomes, particularly hypomethylation, which can have an impact on ORs expression. Importantly, in the manuscript we demonstrate no differences in the methylation patterns of ORs compared to a set of randomly selected genomic regions containing non-OR genes. Thus, this suggests that the expression of these ORs is not a byproduct of global DNA hypomethylation but rather a consequence of a regulated process of gene expression in AML. However, conducting an "extended methylation study" is beyond the scope of this manuscript, which already contains seven figures with up to four items each.

The rationale behind the phylogenetic tree was not clear until the drug exposure section.   Perhaps the authors could comment about the redundancy of the ORs and can they compensate for others?

Thank you for this suggestion. As suggested, we have included a comment regarding the putative functional redundancy of these phylogenetically close ORs. 

Manuscript, page 9 (reviewed version): 

"Interestingly, we confirmed that ORs closely related in the evolutionary phylogenetic tree, such as OR genes from subfamilies 2 and 10, exhibited positively correlated expression patterns (rho > 0.5 and p value<0.05) (Figure 3B), suggesting a potential functional redundancy."

The authors in Figure 6C showed some correlation with some relevant cancer genes, however, it would be useful to show if the differences in outcome shown were related to any specific gene or groups of genes (methylation related for example)

We were unable to understand exactly what this reviewer is suggesting. In Figure 6C, we show the expression correlation of ORs from families 2 (OR2), 13 (OR13), and 52 (OR52). The genes that correlate with these ORs are proto-oncogenes (in green) and tumor suppressor genes (in red). Further to these genes, we did not find any biases regarding any set of specific genes. Regarding the methylation pattern, we also investigated some of these genes, but we did not find any methylation pattern (data not shown). Additionally, these genes are located on different chromosomes (e.g., Cdk6 - chr 7, GAT2 - chr 3, KMT2A - chr 11, MSI2 - chr 17), which would require a very complex mechanism of methylation regulation.

Furthermore, in an attempt to understand the reviewer's question, we thought that the mentioned "outcome" could be referring to the survivals shown in Figures 5A, B, and C. If this is the case, we understand that these results presented in Figure 5A-C and Figure 6C involve lists of ORs that have a small overlap (2 out of 7 ORs), and therefore, we do not see any meaningful comparison in this context.

Reviewer 2 Report

It is a well designed study.

Introduction is expressing quite well the object of the study.Nevertheless oflactory receptors and AML is not in the field of AML research in recent years. Most literature references were over past years. 

Methods and materials are very well analyzed especially in the molecular setting. Characteristics of the AML patients are not mentioned (types, molecular chareteristics, age, prognostic factors). There is a weak point in the presentation of the AML patients. Additionally the sample of 151 patients is quite low.

Results are very well advocated. The implementation of the study and the molecular analysis are presented  in detail. The figures are explanatory. As I have already mentioned in the methods section, results should be commented regarding AML patients characteristics 

Discussion could be more detailed regarding the literature of AML and oflactory receptors. Otherwise, the oncology relation of oflactory receptors and other types of cancers is already known in cancer research and is referred in discussion section. In my opinion haematological approach in cooparetion with the already existing molecular approach of the topic in this study is what is needed. 

It is a well designed study.

Introduction is expressing quite well the object of the study.Nevertheless oflactory receptors and AML is not in the field of AML research in recent years. Most literature references were over past years. 

Methods and materials are very well analyzed especially in the molecular setting. Characteristics of the AML patients are not mentioned (types, molecular chareteristics, age, prognostic factors). There is a weak point in the presentation of the AML patients. Additionally the sample of 151 patients is quite low.

Results are very well advocated. The implementation of the study and the molecular analysis are presented  in detail. The figures are explanatory. As I have already mentioned in the methods section, results should be commented regarding AML patients characteristics 

Discussion could be more detailed regarding the literature of AML and oflactory receptors. Otherwise, the oncology relation of oflactory receptors and other types of cancers is already known in cancer research and is referred in discussion section. In my opinion haematological approach in cooparetion with the already existing molecular approach of the topic in this study is what is needed. 

Author Response

RESPONSE LETTER:

--------

Reviewer #2

Reviewer's points (in black)

Authors' response (in blue)

Comments and Suggestions for Authors

It is a well designed study.

Thank you for taking the time to review our manuscript. We appreciate your insightful comments and constructive feedback, which have helped us to improve the quality of our work. 

Introduction is expressing quite well the object of the study. Nevertheless oflactory receptors and AML is not in the field of AML research in recent years. Most literature references were over past years. 

We acknowledge that the limited focus on the role of olfactory receptors (ORs) in AML research in recent years can be primarily attributed to the fact that ORs have often been overlooked, not only in AML research but also in various types of cancer genomic and transcriptomic studies. This oversight stems from the prevailing assumption that ORs exclusively belong to the olfactory transduction system. Furthermore, AML is a complex disease, and thus, research efforts have predominantly concentrated on well-established genetic and molecular alterations, such as gene mutations in FLT3, NPM1, and DNMT3A, which have received significant attention. The understanding of ORs in non-olfactory tissues is limited, and the technical challenges associated with OR research are not commonly addressed, which may explain the difficulties encountered in studying ORs in AML, both within Hematology laboratories and Olfactory Neurobiology laboratories. The absence of well-designed studies investigating the role of ORs in AML biology and pathogenesis has created a promising area of research to which we aim to contribute. Our study, for the first time, demonstrates the presence of a specific repertoire of ORs expressed in AML cells from patients and leukemic cell lines. Furthermore, we have identified an OR signature with prognostic potential for AML patients. We believe that this study sheds light on this topic and paves the way for additional opportunities and funding in this field of investigation.

Methods and materials are very well analyzed especially in the molecular setting. Characteristics of the AML patients are not mentioned (types, molecular chareteristics, age, prognostic factors). There is a weak point in the presentation of the AML patients. Additionally the sample of 151 patients is quite low.

We appreciate this reviewer for the excellent suggestion. As proposed, we have created a table containing various characteristics related to the samples used in this study (from TCGA and Beat AML). These data have been added to Supplementary Table S1, providing key and important patient characteristics (see next point - below).

Regarding the statement that our set of 151 patients is small, we respectfully disagree with this reviewer. Obtaining a set of 151 patients with gene expression (RNA-seq), exome sequencing, methylation data, and clinical information is a challenging task that requires a well-coordinated team and substantial resources. Furthermore, all our findings have statistical support with a maximum alpha error of 5%. We have also corrected for multiple testing where necessary. While this set of 151 samples may be considered small for investigating factors related to GWS, that was not addressed in our manuscript. Therefore, we believe our results are reliable and robust.

Results are very well advocated. The implementation of the study and the molecular analysis are presented  in detail. The figures are explanatory. As I have already mentioned in the methods section, results should be commented regarding AML patients characteristics 

Thank you for these positive comments regarding our results and figures. Once again, we appreciate this suggestion regarding the patients' characteristics used in this study. As suggested, we have included such information in a supplementary table (Supplementary Table X) and we are also partially describing these patients characteristics in the Results section (page 6 - second paragraph): 

"To find ORs that are over (or majorly) expressed in AML, we evaluated 151 samples from The Cancer Genome Atlas (TCGA). In this cohort, individuals present a median age of 56 years at diagnosis, being 55% of patients male. Most individuals (58.3%) present a FAB classification ranging from M0 to M2, and median percentages of bone marrow and peripheral blood blasts are 72% and 39%, respectively (Supplementary Table S1)."

Discussion could be more detailed regarding the literature of AML and oflactory receptors. Otherwise, the oncology relation of oflactory receptors and other types of cancers is already known in cancer research and is referred in discussion section. In my opinion haematological approach in cooparetion with the already existing molecular approach of the topic in this study is what is needed. 

The suggested adjustments have been made directly in the Discussion section, and additional references as well as supplementary information have been included, as recommended by the reviewer. Please check the manuscript, specifically pages 16-19, to see the implemented changes.

Round 2

Reviewer 1 Report

The authors have responded to my previous comments.   Whilst I recognise that the methylation studies have clarified that the expression of ORs is not related to global DNA hypomethylation - my feeling is that this could have been expanded.   But not a point I feel strong enough to need further revision.   

My point on relating outcome to a specific gene or groups of genes was perhaps not clearly explained as it was not in relation to the OR genes but in terms of the AML related mutations.   Whilst you mentioned that you did this in terms of proto-oncogenes and tumour suppressors this doesn't map to to genes such as "chromatin modifers", "transcription factors" or "signal transduction"  groups of mutated genes.   

Reviewer 2 Report

It is a very good research study. I am not sure if the field of AML is the appropriate for oflactory receptors. The characteristics of AML patients were added but not in a detailed setting. Discussion was corrected and sections were added but not in a point that convince the necessity of the study at present and future of AML research

Quality of English language is good